# Sub-Pixel Convolutional Neural Network for Image Super-Resolution Reconstruction

**Guifang Shao** [1] ⬤, **Qiao Sun** [2] ⬤, **Yunlong Gao** [1], **Qingyuan Zhu** [1,*] ⬤, **Fengqiang Gao** [1] and **Junfa Zhang** [1]

[1] Pen-Tung Sah Institute of Micro-Nano Science and Technology, Xiamen University, Xiamen 361005, China; gfshao@xmu.edu.cn (G.S.); gaoyl@xmu.edu.cn (Y.G.); gaofq@xujc.com (F.G.); s_bblove@sina.com (J.Z.)

[2] School of Engineering, University of Calgary, Calgary, AB T2N 1N4, Canada

[*] Correspondence: zhuqy@xmu.edu.cn

**Abstract:** Image super-resolution (SR) reconstruction technology can improve the quality of low-resolution (LR) images. There are many available deep learning networks different from traditional machine learning algorithms. However, these networks are usually prone to poor performance on complex computation, vanishing gradients, and loss of useful information. In this work, we propose a sub-pixel convolutional neural network (SPCNN) for image SR reconstruction. First, to reduce the strong correlation, the RGB mode was translated into $YC_bC_r$ mode, and the Y channel data was chosen as the input LR image. Meanwhile, the LR image was chosen as the network input to reduce computation instead of the interpolation reconstructed image as used in the super-resolution convolutional neural network (SRCNN). Then, two convolution layers were built to obtain more features, and four non-linear mapping layers were used to achieve different level features. Furthermore, the residual network was introduced to transfer the feature information from the lower layer to the higher layer to avoid the gradient explosion or vanishing gradient phenomenon. Finally, the sub-pixel convolution layer based on up-sampling was designed to reduce the reconstruction time. Experiments on three different data sets proved that the proposed SPCNN performs superiorly to the Bicubic, sparsity constraint super-resolution (SCSR), anchored neighborhood regression (ANR), and SRCNN methods on reconstruction precision and time consumption.

**Keywords:** image super-resolution; convolutional neural network; sub-pixel; residual network

## 1. Introduction

Images, as one of the important media to transmit information, are widely used in industrial production, biomedical, robot vision, and other fields. Image processing is complex and involves various devices. As a result, the images obtained from practical applications are usually of poor quality, with problems such as low resolution, blur, distortion, noise, and loss of detail. To generate a single high-quality and high-resolution (HR) image, the image super-resolution (SR) reconstruction technology is proposed by using multiple low-quality and low-resolution (LR) images with complementary information. It has been widely applied in remote sensing [1], medical imaging [2], and monitoring [3].

In the past decades, many SR methods have been proposed, and they can be divided into four categories according to the interpolation, reconstruction, and machine learning or deep learning methods, as shown in Figure 1.

The interpolation-based method utilizes the correlation between adjacent pixels in the spatial domain, including the nearest neighbor interpolation, the bilinear interpolation, the bicubic interpolation, the multi-surface fitting, and the displacement field-based up-sampling algorithm [4]. These methods are fast but prone to distortion in the edge regions owing to a lack of useful information. To recover the lost high-frequency information, the reconstruction-based algorithms tend to model the imaging process from multiple LR images, but the entire solution process is an ill-conditioned inverse problem. There are

two typical categories in the reconstruction-based methods, that is, the frequency domain algorithms and the spatial domain ones. The former performs worse than the latter due to a lack of prior knowledge. Typical algorithms in the latter include the non-uniform interpolation method (NUI) [5], projection onto convex sets (POCS) [6], iterative back projection (IBP) [7], and maximum a posteriori (MAP) [8]. However, the reconstruction-based methods are too complex to be applied in real applications. To solve the amplified factor restriction and textural details preservation problems in the mentioned methods, machine learning algorithms are proposed to learn the mapping relationship between the HR and LR images. Support vector regression [9], sparse representation [10], and anchored neighborhood regression (ANR) [11] are popular methods in this category. However, a great number of parameters need to be adjusted manually, and the model is also complex, resulting in weak generalization ability.

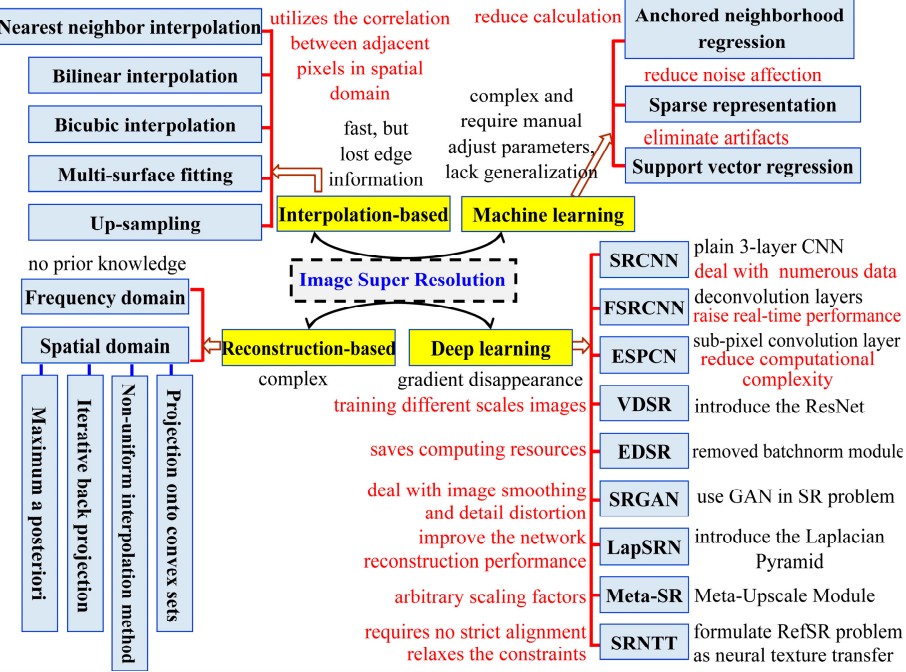

**Figure 1.** Different categories of image SR methods.

Compared with machine learning, deep learning can not only effectively deal with large amounts of data but also reduces the number of parameters that need to be manually adjusted in neural network training, thus remarkably improving the generalization ability. With the development of deep learning, different networks have been put forward to reconstruct SR images. A three-layer structure network using deep convolutional neural networks (SRCNN) was first applied in image SR reconstruction based on feature extraction-nonlinear mapping-image reconstruction [12]. However, there are still some shortcomings, such as loss of high-frequency detail information, blurring of textural details, and low real-time efficiency. To improve the real-time performance deficiency of SRCNN, the faster SR convolutional neural network (FSRCNN), an end-to-end reconstruction network based on the original image, was proposed by introducing a deconvolution layer, reconstructing the mapping layer, using a smaller size filter and adding more mapping layers [13]. Although FSRCNN owns a faster training speed, it has a slow convergence rate and is a shallow neural network. Meanwhile, an efficient sub-pixel convolutional neural network (ESPCN) was put forward to deal with the computational complexity of SRCNN in which the convolution operation is performed on an LR image [14]. In addition, to train different scale images together, a very deep convolutional network (VDSR) was proposed by improving the VGG convolutional network based on ResNet [15,16]. At the same time, a deep recurrent neural network (DRCN) was put forward to deepen the number of network layers, and it

achieves better reconstruction results than SRCNN with the same weight parameters in the convolutional layer of the recursive loop [17]. As the ResNet is not suitable for low-level computer vision problems, an enhanced deep SR (EDSR) was proposed by utilizing an enhanced ResNet and training with L1 regular loss function [18]. EDSR saves computing resources by removing the unnecessary batch normalization layer in the original ResNet and obtains better training results. Furthermore, to solve the image smoothing and detail distortion problems arising from magnifying images too many times, the SR reconstruction based on a generative adversarial network (SRGAN) uses a perception loss instead of minimum mean square error (MSE) in the traditional methods [19]. To further improve the network reconstruction performance, a cascade structure and a new Charbonnier loss function are combined in the Laplacian pyramid SR network (LapSRN) to gradually optimize the LR image [20]. In addition, to solve the arbitrary scaling factors in most existing SR methods, the Meta-SR was put forward by introducing a meta-upscale module to replace the traditional upscale one [21]. At the same time, to deal with the problem of false textural and artificial features in SR methods, SR neural texture transfer (SRNTT) was proposed by formulating the reference-based SR (RefSR) problem as neural texture transfer and by performing multi-level matching in neural space, so as to promote multi-scale neural transmission [22]. The advantage of SRNTT is that there is no strict alignment requirement between the reference image and the blurred image, and the constraints on the existing RefSR method are also relaxed. Then, in agriculture, DAFTGAN [23] was proposed, which combines DenseNet [24] and ResNet in a single layer, significantly reducing the number of parameters used in training the deeper network structure. To utilize feature information at different levels and deal with the gradient disappearance problem, a multi-hierarchical features fusion network (MHFFN), by introducing a dual residual block, is designed for single image SR [25].

In summary, although the existing deep learning-based image SR reconstruction methods already show improvements in calculation efficiency [13,14,16,17], support multi-scale training [15,20,21], and better restore textural details [18,19,21], there still exist difficulties in vanishing gradients, gradient explosion, complex computation, and other problems as the network layers deepen. More importantly, the LR image in the mentioned networks is first amplified to an HR image with the same size as the target SR image, which will increase the computational complexity.

In this paper, we propose a sub-pixel convolutional neural network (SPCNN) for SR image reconstruction. First, the LR image is used as the network input to reduce the time consumed by image amplification in the image preprocessing steps of SRCNN and DRCN. Then, we build two convolutional layers as feature extraction layers to obtain more information. Meanwhile, to achieve more features on different levels, four non-linear mapping layers are designed. Subsequently, to reuse feature information and avoid vanishing or exploding gradients arising from network layer deepening, the Residual network is introduced with a shortcut connection to directly transfer information from lower network layers to higher layers. Finally, to reduce image reconstruction time and maintain the feature correlation unchanged, we design a sub-pixel convolutional layer based on up-sampling to reconstruct the HR image, which rearranges the same information in multiple feature images according to pixel.

This article is organized as follows. Section 2 describes the proposed method and provides more detail. Comparative experiments on different data sets and discussions are presented in Section 3. The main conclusion is summarized in Section 4.

## 2. Method

Inspired by SRCNN with three layers and DRCN with deep layers, this paper proposes an image SR reconstruction method based on sub-pixel convolutional neural networks named SPCNN. There are five main parts to our proposed method, as shown in Figure 2.

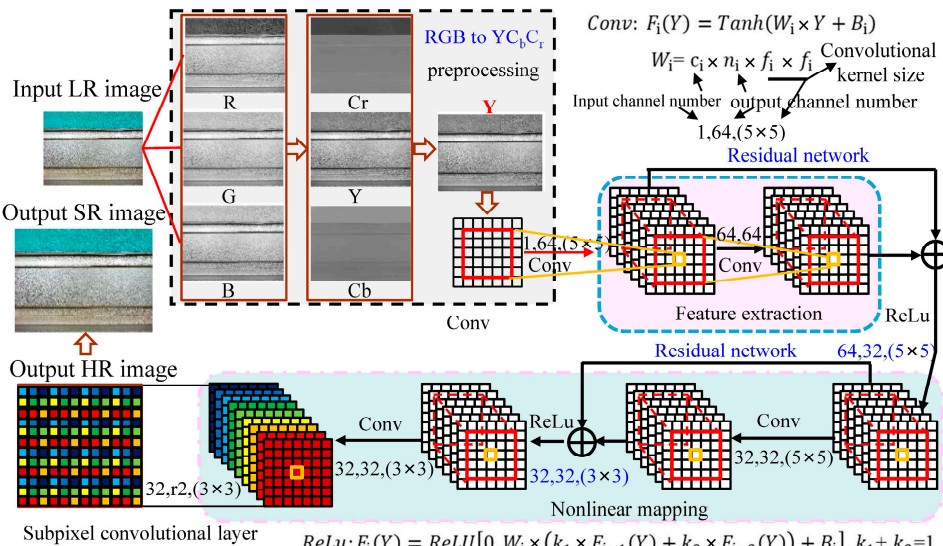

**Figure 2.** The flowchart of the proposed SPCNN.

First, the LR image is adopted as the network input and is transformed into $YC_bC_r$ mode. Then, two convolutional layers for feature extraction and four nonlinear mapping layers for achieving more features are designed. Furthermore, the residual network is introduced to maintain the features in lower layers. Finally, a sub-pixel convolutional layer is put forward to obtain the HR image from the LR image feature space. Therefore, the proposed network contains a total of 7 layers, and each layer is defined as Conv (input, output, filter), in which input is the number of input channels; output denotes the number of output channels, and filter represents the size of the convolution kernel.

### 2.1. Image Preprocessing

The enlarged image instead of the LR image is used in SRCNN and DRCN, which will require more convolution calculations during the network training process. Therefore, to reduce the time consumed by convolution calculations, the LR image is directly adopted as input in our method. In addition, as the LR image is colorful in RGB mode and there is a strong correlation in the spatial domain, here we first transform the image from RGB mode into $YC_bC_r$ mode. More importantly, we choose the Y channel image as the final network input according to experimental analysis showing that it can achieve similar reconstruction results to that of $YC_bC_r$ [26].

### 2.2. The Network Structure

To obtain more feature information, two convolutional layers, Conv1 and Conv2, are constructed in the feature extraction stage, as shown in Figure 3. Here, 64 convolution kernels of $5 \times 5$ are employed in both Conv1 and Conv2 to ensure the low-dimensional feature information is as rich as possible. The network takes the Y channel image as the input of Conv1, and conducts feature extraction operations with 64 convolution kernels of $5 \times 5$ to obtain a low-dimensional feature image set $F_1(Y)$ by the following formula

$$F_i(Y) = Tanh(W_i \times Y + B_i), \tag{1}$$

in which, $W_i$ and $B_i$ are the weights and biases of the network, respectively. The size of $W_i$ is $c_i \times n_i \times f_i \times f_i$, with $c_i$ representing the number of channels of the input image; $n_i$ is the number of network convolution kernels, i.e., the number of output channels; $f_i \times f_i$ denotes the size of the convolution kernel, and $Tanh(x)$ is the activation function and is used in all convolutional layers for activation. Similarly, let $F_1(Y)$ be the input of Conv2 and feature image set $F_2(Y)$ can be achieved by operating with 64 convolution kernels of $5 \times 5$, to provide supplementary information for the reconstruction of the subsequent HR image.

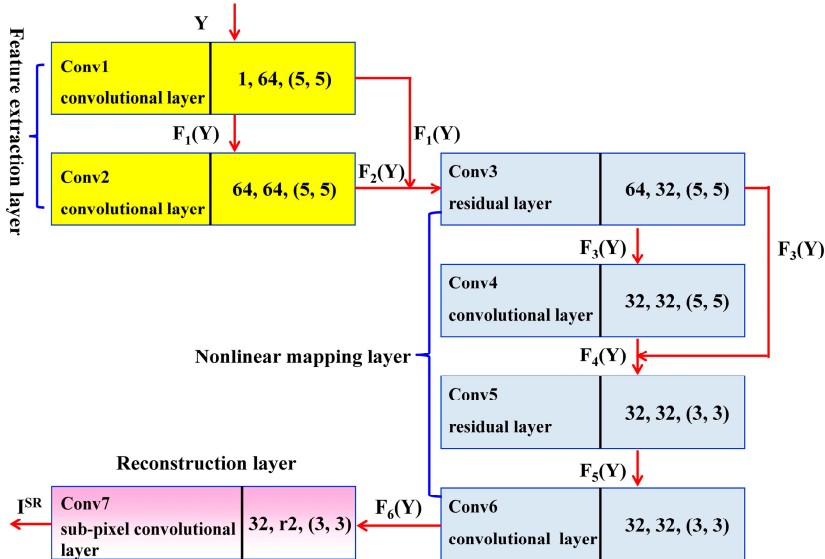

**Figure 3.** The proposed network and parameter setup.

The Conv3 layer takes the feature image set $F_1(Y)$ as the identity mapping $F_2(Y)$ as the convolution branch input, and performs the convolution calculation with 32 convolution kernels each of size $5 \times 5$, to obtain the feature image set $F_3(Y)$. Conv4 takes $F_3(Y)$ as input and performs a second mapping with 32 convolution kernels of size $5 \times 5$ to obtain the feature image set $F_4(Y)$. Furthermore, the remaining layer Conv5 takes $F_3(Y)$ and $F_4(Y)$ as input and performs a third mapping with 32 convolution kernels of size $3 \times 3$ to obtain a feature image set with higher dimensions and richer texture details $F_5(Y)$. Conv6 takes $F_5(Y)$ as input and performs the last non-linear mapping with 32 convolution kernels of $3 \times 3$ to obtain a high-dimensional global feature map set $F_6(Y)$. The operations of Conv4 and Conv6 are similar to Conv1, described as Formula (1). The specific operations of Conv3 and Conv5 are shown by the following formula

$$F_i(Y) = ReLu[0, W_i \times (k_1 \times F_{i-1}(Y) + k_2 \times F_{i-2}(Y)) + B_i)], \tag{2}$$

in which $i$ is the layer number, and $k_1$ and $k_2$ denote the proportion of $i - 1$ and $i - 2$ layers feature maps in this layer image channel. To maintain data consistency, let $k_1 + k_2 = 1$. The size of $W_i$ is $(k_1 \times F_{i-1}(Y) + k_2 \times F_{i-2}(Y)) \times n_i \times f_i \times f_i$, and $Re(x)$ is the activation function.

To further obtain the final HR image, the sub-pixel convolution layer Conv7 is introduced to reconstruct the HR images. In Conv7, the global feature map set $F_6(Y)$ is used for reconstruction to obtain the final HR image $I^{SR}$. In fact, the number of image channels becomes $r^2$ after being processed by the convolution layer and the remaining layer, and these images can be rearranged in a sub-pixel convolution layer to complete the image reconstruction. The specific operation is shown by the following formula

$$I^{SR} = PS(W_i \times F_{i-1}(Y) + B_i), \tag{3}$$

in which the size of $W_i$ is $r^2 \cdot c \times n_i \times f_i \times f_i$ and $c$ is the number of input channels of the initial LR image. $PS$ denotes the sub-pixel convolution operation, meaning that the channel image is rearranged according to pixel position, and the image features are integrated. $I^{SR}$ represents an HR image after reconstruction.

### 2.3. Remaining Network

Generally, deepening the network structure is a strategy to improve the reconstruction performance of deep learning-based methods. As the number of network layers rises, richer feature information can be extracted, and the deep network can extract some abstract information that the lower-layer network cannot obtain. However, the increasing depth of

the network will cause vanishing or exploding gradients, and the model will degenerate into an identity mapping, that is, the deep network will degenerate into a shallow network. Therefore, the remaining network structure is introduced into the nonlinear mapping stage to deepen the network and to improve the reconstruction performance. The remaining network includes a convolution layer with a shortcut connection and ReLU activation function, as shown in Figure 2. In the residual network structure, the low-level network input $Y$ is directly connected with the output layer through a shortcut connection, so that the output result becomes $(Y) = F(Y) + Y$ and reuses feature information. When $(Y) = 0$, then $H(Y) = Y$, which is an identity mapping, so as to ensure no decrease in network accuracy with the increased depth.

### 2.4. Sub-Pixel Convolutional Layer

In SRCNN, the initial LR image is first interpolated by the bicubic algorithm, which corresponds to a down-sampling operation on the reconstructed HR image. Assuming that the sampling factor is r and c denotes the number of color channels, let $H \times W \times c$, $rH \times rW \times c$ and $rH \times rW \times c$ represent the dimensions of the real tensor of the initial LR image, the LR image after bicubic interpolation, and the reconstructed HR image, respectively. Evidently, the LR image after bicubic interpolation has the same tensor dimensions as that of the reconstructed HR image, and it is extremely time-consuming. Therefore, a sub-pixel convolutional layer is introduced to implement SR image reconstruction by performing an up-sampling operation in the output layer of the neural network.

For the sub-pixel convolution layer, its input is the feature map with $r^2$ channels, and $r$ denotes the up-sampling factor. By rearranging the $r^2$ channels of each pixel into the image area with size $r \times r$, the LR feature images with real tensor dimensions $H \times W \times C \cdot r^2$ are transformed into an HR image of $rH \times rW \times C$. Therefore, to achieve the image reconstruction effect, the sub-pixel convolutional layer aims to rearrange the LR feature vector in dimension $r$ according to certain rules. For example, if the operation of the sub-pixel convolution layer with $r = 2$, where the pixels at the same position in the feature image are rearranged and integrated to form a corresponding block in the HR image. Here, every 4 pixels form an area.

To be specific, let the LR image feature space conduct a convolution operation with a filter $W_s$ in size $f_s$ and a weight interval of $\frac{1}{r}$. Partial weights in the filter $W_s$ will be activated, yet other weights of each pixel that are not involved in the calculation will not be activated. When the number of activations is $r^2$, there are at most $\left\lceil \frac{k_s}{r} \right\rceil^2$ weights that are activated according to the position distribution of the model. During the convolution operation, the filter will scan the image according to the position of the sub-pixel, and the scanned weights will be activated. Let $mod(x, r)$, $mod(y, r)$ be the horizontal coordinate $x$ and vertical coordinate $y$ of an output pixel in HR image feature space. The sub-pixel convolution operation, an actual up-sampling operation when $mod(f_s, r) = 0$, can be implemented as follows

$$F_L(Y) = PS(W_L \times Y + b_L), \tag{4}$$

$$PS(T)_{x,y,c} = T_{\lfloor x/r \rfloor, \lfloor y/r \rfloor, C \cdot r \cdot mod(y,r) + C \cdot mod(x,r) + c}, \tag{5}$$

in which $Y$ is an LR feature image with real tensor dimensions $H \times W \times C \cdot r^2$; the size of $W_L$ is $n_{L-1} \times r^2 C \times f_L \times f_L$, and $PS$ $(T)$ denotes the periodic permutation operator, i.e., the sub-pixel convolution operation.

When $f_L = \frac{f_s}{r}$ and $mod(f_L, r) = 0$ are satisfied, the sub-pixel convolution operation can be performed in LR image feature space with the filter $W_s$.

## 3. Experiments and Discussion

### 3.1. Quantitative Evaluation

The purpose of SR reconstruction is to improve the image resolution as large as possible, at the same time keeping the image details as clear as possible. Thus, it is essential

to select appropriate evaluation indexes for the reconstructed image. To quantitatively evaluate the reconstructed image and the original image, some commonly used methods, including the structural similarity (SSIM) [27] and peak signal-to-noise ratio (PSNR) [28], are utilized.

The SSIM is designed to measure the image brightness, contrast, and structure. Assuming that the original image is $x$ and the reconstructed image is $y$, the direct SSIM between $x$ and $y$ can be derived from the following formulas

$$\text{SSIM}(x, y) = \frac{\left(2\mu_x\mu_y + c_1\right)\left(2\sigma_{xy} + c_2\right)}{\left(\mu_x^2 + \mu_y^2 + c_1\right)\left(\sigma_x^2 + \sigma_y^2 + c_2\right)}, \tag{6}$$

$$c_1 = (k_1 \times L)^2, \tag{7}$$

$$c_2 = (k_2 \times L)^2, \tag{8}$$

in which, $\mu_x$ and $\mu_y$ are the average brightness of image $x$ and y, respectively. $\sigma_x$ and $\sigma_y$ are the variances of images $x$ and $y$, referring to the estimation of image contrast. $\sigma_{xy}$ represents the covariance between image $x$ and $y$, meaning the structure of the estimated image. $c_1$ and $c_2$ are fixed values; $k = 0.01$; $k = 0.03$; $L = 255$. The value of SSIM ranges from 0 to 1. The closer the value of SSIM is to 1, the more the reconstructed image is similar to the original image; that is, a better reconstruction is obtained.

The quality of the reconstructed image can be evaluated by calculating the error between the pixels of the image. PSNR can be obtained from the following formulas:

$$MSE = \frac{\sum_{i=1}^{W}\sum_{j=1}^{H}[X(i,j) - Y(i,j)]^2}{W \times H}, \tag{9}$$

$$\text{PSNR} = \log_{10}\left[\frac{(2^n - 1)^2}{MSE}\right], \tag{10}$$

in which, $X$, $Y$ are the pixel matrix of the original image and the reconstructed HR image, respectively. $MSE$ denotes the mean square error between $X$ and $Y$. $H$ and $W$ are the height and width of the image. $n$ is the number of bits for each pixel. Here, the value of $n$ is 8. $X(i, j)$ and $Y(i, j)$ represent the desired real image and reconstructed image, respectively. The larger value of PSNR, the smaller degree of distortion, the better the image reconstruction.

In addition, to verify the effectiveness of the model proposed in this article, Bicubic [29], SCSR [30], ANR [11], and SRCNN [12] are chosen to compare different data sets.

### 3.2. Experiment Setup and Data Sets

To effectively verify the performance of SPCNN proposed in this article, the NVIDIA graphics card GeForce GTX 1050 Ti is used to build an experimental environment under the Ubuntu 16.04 operating system. The detailed configuration is shown in Table 1.

**Table 1.** Experimental hardware environment configuration.

| Name | Description |
| --- | --- |
| computer model | Dell-Optiplex-3020 |
| graphics card | GeForce GTX 1050 Ti (4 GB) |
| RAM | 8 GB |
| processor | i3-4160 CPU |
| operating system | Ubuntu 16.04 (64 bit) |
| main frequency | 3.60 Ghz |

Here, the initial values of relevant parameters in the network are set, as shown in Table 2.

**Table 2.** Setup of parameters in the network.

| Parameter | Description | Initial Value |
|---|---|---|
| batch_size | every grouped data sets for training | 64 |
| epoch | the maximum iterations of algorithm | 100 |
| Lr | the initial learning rate used in Adam [29] to update the weight matrix | 0.01 |
| gamma | the learning rate attenuation factor in MultiStepLR learning strategy | 0.1 |

### 3.3. Comparison of Public Data Sets

To ensure the reliability of comparative experiments, the public data set VOC2012 is selected as training samples. Here, 16,700 images are used as training data and 425 images as testing data. Table 3 shows the evaluation indicators of five methods on data Set5.

**Table 3.** PSNR and SSIM comparison on data set5.

| Images | Bicubic | SCSR | ANR | SRCNN | SPCNN | Bicubic | SCSR | ANR | SRCNN | SPCNN |
|---|---|---|---|---|---|---|---|---|---|---|
| | | | **PSNR** | | | | | **SSIM** | | |
| baby | 33.19 | 34.29 | 35.13 | 35.01 | **35.21** | 0.903 | 0.904 | 0.902 | 0.921 | **0.932** |
| bird | 32.58 | 34.11 | 34.60 | 34.91 | **35.11** | 0.926 | 0.939 | 0.949 | 0.949 | **0.953** |
| butterfly | 24.04 | 25.58 | 25.90 | 27.58 | **27.63** | 0.824 | 0.863 | 0.872 | 0.889 | **0.901** |
| head | 32.88 | 33.17 | 33.63 | 33.55 | **33.81** | 0.799 | 0.802 | 0.823 | 0.823 | **0.842** |
| woman | 28.56 | 29.94 | 30.33 | 30.92 | **31.06** | 0.891 | 0.905 | 0.917 | 0.923 | **0.933** |
| Ave | 30.39 | 31.42 | 31.92 | 32.39 | **32.56** | 0.869 | 0.883 | 0.897 | 0.901 | **0.912** |

According to the data in Table 3, the PSNR and SSIM of SPCNN are overall higher than other algorithms. As for the PSNR, the average increases are 2.17 dB, 1.14 dB, 0.64 dB, and 0.17 dB, respectively, for an average increase of about 1.03 dB. For the SSIM index, there are smaller improvements of 0.043, 0.029, 0.015, and 0.011, respectively, for an average increase of about 0.0245.

Figure 4 illustrates the data in Table 3. Evidently, there is a great difference in the reconstruction effect of various images, but a similar trend for different methods can be observed. The dashed lines prove that SPCNN performs better than the other four methods, with a smaller increase in average PSNR and a larger growth in average SSIM value. As the image number in Set5 is limited, the experimental results may be unable to verify the effectiveness of SPCNN. Therefore, data Set14 is selected for an in-depth comparison.

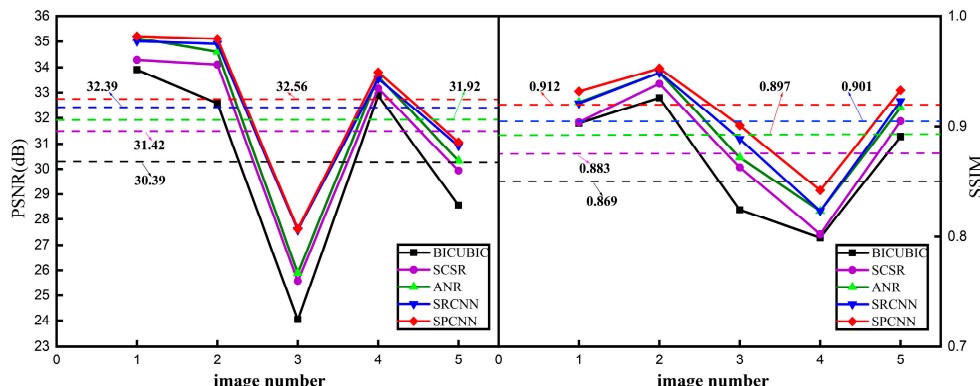

**Figure 4.** Schematic diagram of PSNR and SSIM of data Set5 (dashed lines are mean values).

Figure 5 exhibits the quantitative comparison of PSNR and SSIM on data Set14. As shown in Figure 5, compared to the other four methods, the average PSNR of SPCNN is increased by 1.6 dB, 0.83 dB, 0.53 dB, and 0.14 dB, respectively, for an average increase of about 0.775 dB. Meanwhile, SSIM increased by 0.062, 0.040, 0.027, and 0.013, respectively,

with an average increase of about 0.0355. Although the average increase in the PSNR value in Set14 is lower than that in Set 5, the average SSIM index has higher growth. There is also a great change in image reconstruction owing to the greatly varying texture detail for different data. For images with numbers 8 and 9, due to the robustness of SPCNN and SRCNN compared with other algorithms, the change in their PSNR is quite gentle. As can be seen from the dashed line, the average PSNR value and SSIM value of SPCNN are greater than those of other algorithms.

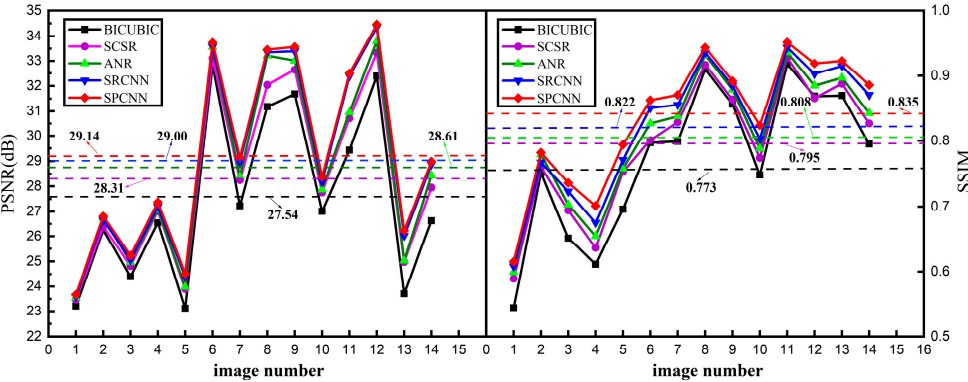

**Figure 5.** Schematic diagram of PSNR and SSIM of data Set14 (dashed lines are mean values).

To further verify the reconstruction effect of our proposed method, Figure 6 displays the results of five algorithms on images selected from data Set5 and data Set14. More importantly, a partial detail is shown in amplification to clarify the reconstruction effect.

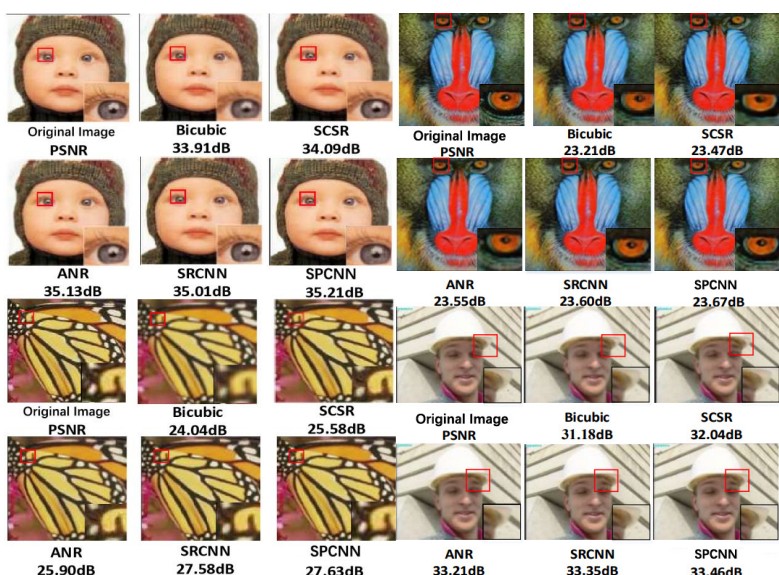

**Figure 6.** Comparison of reconstruction effects on different data sets. Left are Baby and Butterfly from data Set5; right are Baboon and Foreman from data Set14.

It can be seen from the left images in Figure 6 that there is a similar reconstruction effect of the baby image for Bicubic and SCSR, but an obvious artifact phenomenon occurred for the Bicubic algorithm in the eye details, indicating that its reconstruction is worse than that of SCSR. Meanwhile, the detailed reconstruction result of ANR is closer to that of SRCNN, but the PSNR value of ANR is slightly higher. As for SPCNN, it performs best, and the reconstruction of eye detail is closer to the original image. Similarly, a serious artifact phenomenon happened on the Butterfly image with the Bicubic method. SCSR, ANR, and SRCNN have similar effects with the loss of some details. Especially there is an

expansion in the bright white spot area compared with the original image. SPCNN retains better-detailed information in the spot area and is closer to the original image.

One can see from the right images in Figure 6 that there is a similar blurred phenomenon to Mosaic in the eye part of the Baboon image. Compared with the Bicubic method, the reconstruction effect of SCSR, ANR, and SRCNN is improved. However, SCSR and ANR lose more serious eye structural features, yet SRCNN can retain better structural features. As for SPCNN, the structural features are better maintained, and there is no obvious loss of textural detail. As there is less color feature and texture detail in the foreman image, the reconstruction effect of different algorithms is relatively close, but the fuzzy phenomenon still exists in the Bicubic method. SPCNN can retain more detailed features on various images.

In summary, the Bicubic algorithm performs worst on image reconstruction with texture detail loss and artifacts. SCSR and ANR have similar reconstruction abilities, but edge information loss still exists. The overall reconstruction ability of SRCNN is improved on all images, but SPCNN performs better in texture details, and there is no obvious artifact phenomenon.

Figure 7 compares the time consumption of different algorithms on data Set5 and Set14. One can see from Figure 7 that the average reconstruction time of SCSR is the longest, with 35.92 s and 84.88 s on data Set5 and Set14, respectively, which cannot meet the requirement of most industrial environments. The time consumption of SPCNN is the shortest, with an average reconstruction time of about 0.09 s on data Set5 and 0.27 s on data Set14, respectively, indicating that the proposed operations, choosing the LR image as input and designing the sub-pixel convolution layer based on up-sampling, are effective for reducing the reconstruction time.

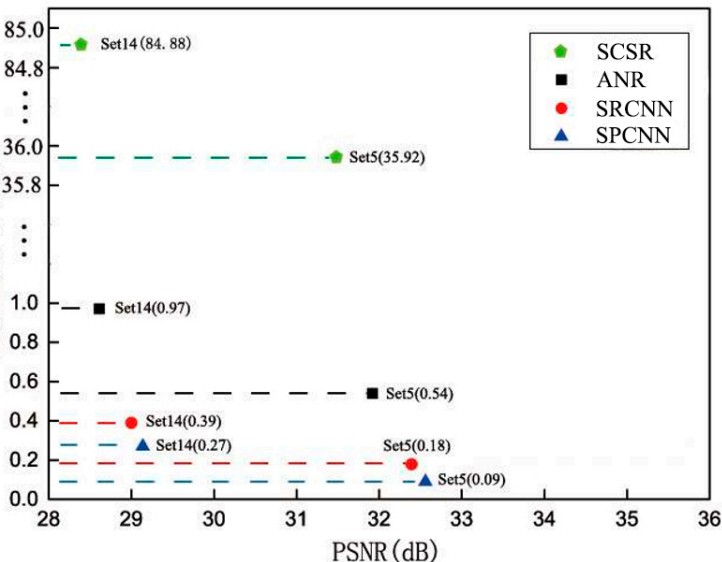

**Figure 7.** The average reconstruction time of different methods on data Set5 and data Set14.

### 3.4. Comparison of Transport Data Set

A private intersection surveillance image traffic data set provided by the Ministry of Transportation of Longhai is used. At the same time, the peak signal-to-noise ratio of PSNR and the structural similarity of SSIM are used to evaluate these methods.

The surveillance video images provided by the Ministry of Communications mainly contain two parts of data, at the entrance of a school and at the intersection of the China Merchants Bureau. For the entrance of the school data, there are various surveillance images taken at four time periods, 6 o'clock, 12 o'clock, 18 o'clock, and 23 o'clock. These images are also affected by different factors, such as the images taken at 6 o'clock are obscured by fog due to rain; images taken at 12 o'clock and 18 o'clock contain different

lighting due to changing cloud formations in the sky, and the detailed information in images taken at 23 o'clock are severely distorted by vehicle lighting. There are 100 images taken at different times and places. Therefore, there is a total of 500 traffic images in the transportation data set.

To conduct the comparison experiments, 450 images are selected as the training set, and the remaining 50 images are used as the testing set. To display the results, three images at different times and places are randomly selected, that is, 6, 12, 18, and 23 o'clock. Figure 8 reveals the comparison results of PSNR and SSIM of five methods on the traffic images.

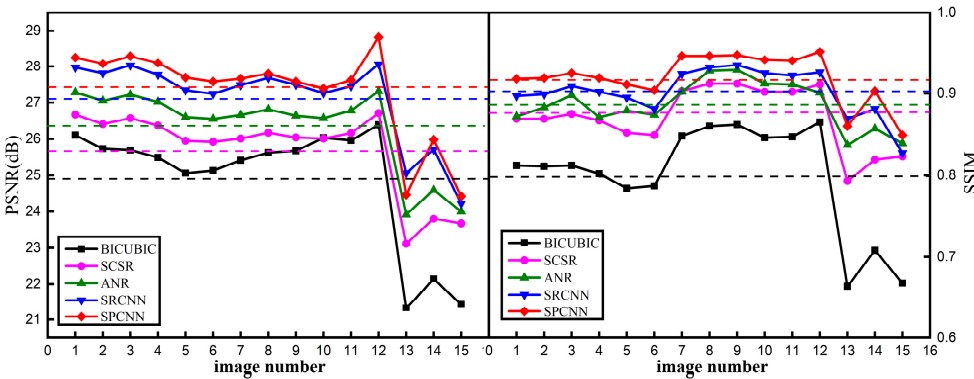

**Figure 8.** Schematic diagram of different methods on traffic data.

As the background characteristics are similar in the transportation data set, there are similar results on images taken at the same place. The proposed SPCNN also performs best on almost all images except image number 13 at the intersection of China Merchants. On average, there is an increase of 1.32 dB on PSNR and 0.055 on SSIM for SPCNN, respectively.

It can be seen from Figure 8 that there are best reconstruction results on images taken at 18 o'clock (image number 7–9) due to less light interference. Meanwhile, as there are fewer vehicles on the road, the reconstruction result of images taken at 23 o'clock (image number 10–12) is similar to that of images taken at 18 o'clock. However, owing to more vehicles in the intersection surveillance image, its reconstruction effect is quite different on various images (numbers 13–15). In general, SPCNN performs best with higher average PSNR and SSIM values.

To further display the reconstruction effect of SPCNN on the traffic data set, Figure 9 illustrates the reconstruction results at different time periods. In Figure 9a, the image reconstruction effect of the Bicubic algorithm is the worst, with obvious blurring in the image. The reconstruction effect of the SCSR algorithm and the ANR algorithm is relatively close, but the details of the vehicle and the background building are lost. There is an obvious image-sharpening phenomenon appearing in SRCNN. The details of SPCNN are relatively complete, but there also exist certain distortions on the sky background. For the image in Figure 9b, as it concerns the vehicles on the road at monitoring, there is less sky detail in the image. Its reconstruction effect is consistent with that in Figure 9a; i.e., the Bicubic algorithm is worst, and SPCNN performs best.

For the surveillance image of the intersection of China Merchants, there are more details than the surveillance image of the school. The reconstruction result of the Bicubic algorithm is still fuzzy; those reconstructed by SCSR and ANR also have artifacts, and SRCNN contains sharpening phenomena. The details of SPCNN are well preserved and are closer to the original image, especially in the text details. It can be seen from the bottom in Figure 9b that the text reconstruction effect of the Bicubic method cannot be effectively recognized. Although the reconstruction result of SCSR and ANR can be recognized, there are serious artifacts. For SRCNN, its reconstruction result is much clearer, and there is a small distortion phenomenon. As for SPCNN, its reconstruction result can be clearly identified and is closer to the original ones.

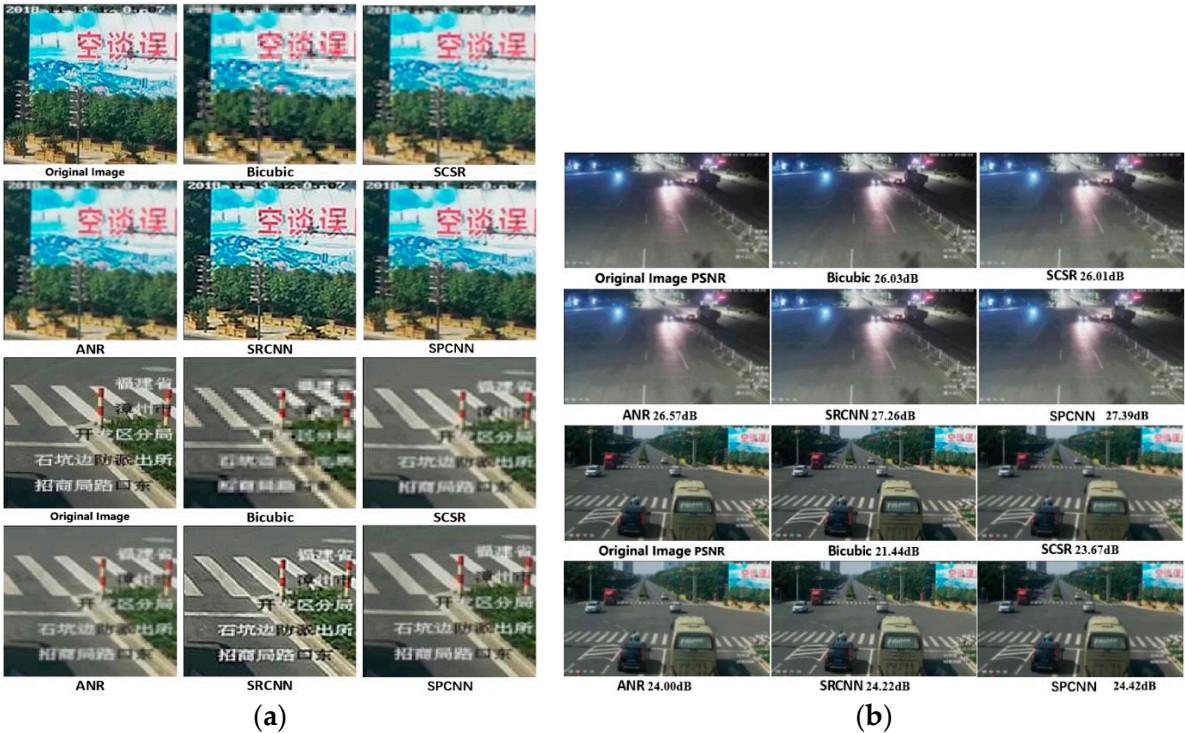

(**a**)                              (**b**)

**Figure 9.** (**a**) China Merchants intersection monitoring image text details. (**b**) Snapshots from top to bottom are the monitoring images of the main entrance of the school at 23 o'clock and the monitoring images of the intersection of China Merchant: China Merchants intersection monitoring image text details.

In summary, the image reconstruction effect of the Bicubic algorithm is the worst; there is an obvious blur phenomenon, and text details are severely lost. The reconstruction effect of the SCSR algorithm and ANR algorithm are relatively close with an edge information loss. Compared with the original image, the overall reconstruction effect of SRCNN is improved, but there is a small amplitude distortion. At last, the reconstruction effect of SPCNN is better in texture details, and there is no obvious artifact.

### 3.5. Comparison of Defect Data Set

Furthermore, these methods are compared to the aluminum profile defect data set. As the industrial defect images are remarkably different from those natural scene images, the network needs to be retrained. Here, 50 defect images are randomly chosen as testing data, and the remaining 450 images are considered as training data. Figure 10 shows the PSNR and SSIM results of five algorithms on images from six different defect types.

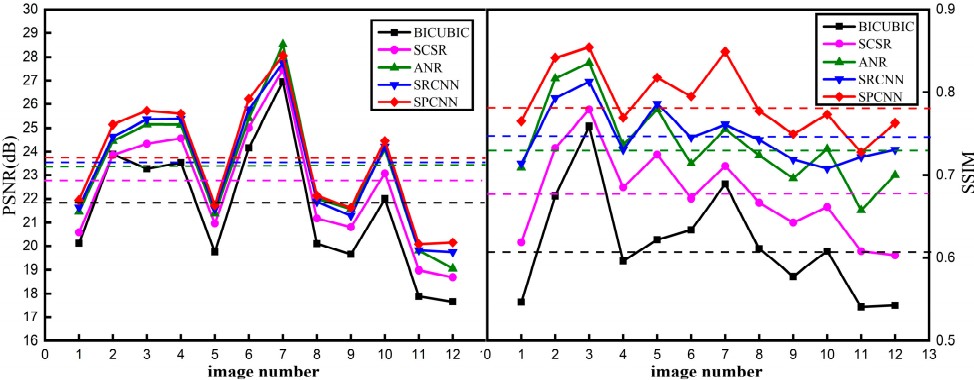

**Figure 10.** Schematic diagram of PSNR and SSIM of Defect data set (dashed line is mean value).

It can be seen from Figure 10 that all the SR reconstruction algorithms do not perform well on the defect data. The SSIM of the Bicubic algorithm only ranges from 0.5 to 0.7. Although the index values of SCSR and ANR algorithms are improved, the PSNR value is still lower than 20 dB, and the SSIM value is lower than 0.7. The SSIM of the SRCNN algorithm is better than other algorithms, basically between 0.7 and 0.8, but some PSNR values are still below 20 dB. The overall value of SPCNN is higher than that of other algorithms. The PSNR values are all higher than 20 dB, and the SSIM value is basically higher than 0.75.

There is a large gap between image reconstruction effects on different defects. There is a significant difference between various algorithms, in which the SSIM values of SRCNN and ANR are close, but SRCNN is worse than ANR in some images. As the dashed line indicates, although the PSNR of SPCNN increases slowly, its reconstruction effect is significantly improved on the SSIM value. Therefore, it means that the SPCNN contains better adaptability to industrial application scenes.

To further verify the effectiveness of SPCNN, six different types of defect images are selected for a comparative experiment, as shown in Figure 11. Obviously, these defect images may contain a great number of bright spot noises, yet the image texture features are less visible. The Bicubic algorithm performs worst with strong blur in the reconstruction image. The reconstruction effect of SCSR and ANR is similar, but there are still some artifacts. SRCNN performs better than SCSR, ANR, and Bicubic, but there is a sharpening phenomenon. The reconstruction effect of SPCNN is slightly better than that of SRCNN, which is close to the original image.

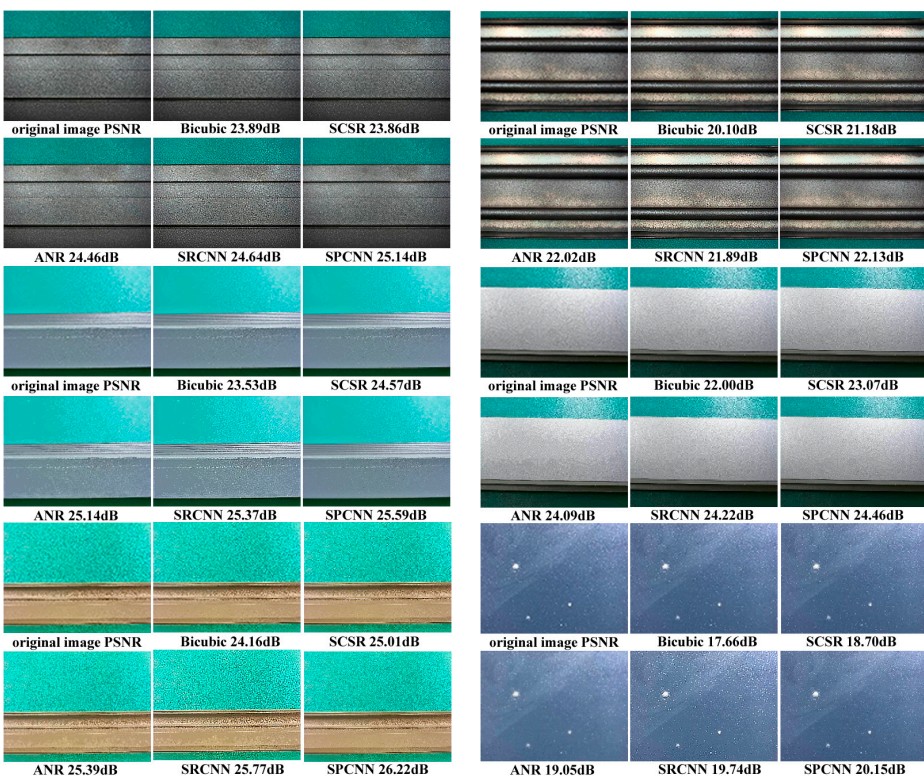

**Figure 11.** Reconstruction effect of different methods on Defect data set. (**Left**) Snapshots from top to bottom are non-conductive defects, scratches, and leakage defects. (**Right**) Snapshots from bottom to top are bottom leak defect, jet defect, and dirty spot defect.

## 4. Conclusions

Image reconstruction has been widely used in remote sensing imaging, medical imaging, and public safety. To deal with the computational complexity, vanishing and exploding gradients, and other problems appearing in current deep learning networks, we proposed

the SPCNN for image SR. At first, the luminance channel of the low-resolution image is taken as the network input directly to simplify computation. Then, a two-layer convolutional network is introduced to enrich the information and feature extraction. Furthermore, the residual network structure is used to solve the problem of vanishing and exploding gradients. Finally, a sub-pixel convolutional layer based on an up-sampling operation is proposed to reduce the image reconstruction time while the correlation of feature information remains unchanged. Comparative experiments between the proposed SPCNN, SRCNN, Bicubic, ANR, and SCSR on public benchmark data sets, private transportation and defect data sets are conducted. The PSNR, SSIM, and visual effects reveal that Bicubic performs worst, and SPCNN is the best one; ANR and SCSR are better than Bicubic, and SRCNN is worse than SPCNN.

Our future work involves studying how to carry out SR reconstruction on video, optimizing the neural network model to better extract the image feature information, and generating suitable training data.

**Author Contributions:** Conceptualization, Q.S. and G.S.; methodology, G.S.; software, F.G.; validation, F.G., Y.G. and Q.Z.; formal analysis, Q.S.; investigation, Y.G.; resources, F.G.; data curation, J.Z.; writing—original draft preparation, J.Z.; writing—review and editing, G.S.; visualization, G.S.; supervision, Y.G.; project administration, Q.Z.; funding acquisition, Q.Z. All authors have read and agreed to the published version of the manuscript.

**Funding:** This work was supported by the Natural Science Foundation of Xiamen under Grant 3502Z20227189.

**Acknowledgments:** We thank Y. Y. Wen for his help in the revision of this manuscript.

**Conflicts of Interest:** The authors declare no conflict of interest.

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
