# Peer review of "Sub-Pixel Convolutional Neural Network for Image Super-Resolution Reconstruction"

_electronics, doi:10.3390/electronics12173572_

Round 1
Reviewer 1 Report
A paper about super-resolution reconstruction should present images in a quality that is high enough to assess the quality of reconstruction visually. This requires a ground-truth image (high resolution) an input image (low resolution) and the reconstructed outputs using different state of the art vs the proposed method. This presentation should have the highest possible image quality (i.e. the publication should not have compressed images or images with reduced resolution). If this is not possible due to digital production issues, the raw images should be provided as additional materials.
The paper is easily readable but would profit from a proof-reader that is native in english.
Author Response
We are grateful for the suggestion of the reviewer and provide the comparison as following. Taking the butterfly image in data Set5 as example, Fig.1a illustrates the ground-truth image in size of 256*256, Fig.1b is the input low resolution image in size of 128*128. Fig.1c to Fig.1f are the reconstruction results by different methods, obviously, Bicubic method will blur the image and loss lots detail information. Although the reconstruction results of ANR, SRCNN and SPCNN seems similarly, the middle segmentation line in yellow part (see the red rectangle) of ANR and SRCNN are almost disappeared. The proposed SPCNN can preserve more detail information.

Reviewer 2 Report
This paper proposes a sub‐pixel convolutional neural network (SPCNN) for image SR reconstruction.
The topic is relevant and original. The author began by reducing the strong correlation, the RGB mode was translated into YCbCr mode, and the Y channel data was chosen as the input LR image. The LR image was chosen as the network input to reduce computation instead of the interpolation reconstructed image. Then, two convolution layers were built to obtain more features. Also, four non‐linear mapping layers were used to achieve different level features. As well, the residual network was introduced to transfer the feature information from lower layer to higher layer to avoid the gradient explosion or vanishing gradient phenomenon. Finally, the sub‐pixel convolution layer based on up‐sampling was designed to reduce the reconstruction time.
The topic is relevant and original. The authors made a good introduction concerning the actual research in image SR methods with actual and appropriate references.
The method used is clearly described and comprehensively presented with suitable experiments and discussion.
In conclusions ‘section the authors emphasise their contributions and future possible applications.
The references are appropriate and more than 50% are from the 5 last years.
Final recommendation:
The subject is well within the scope of the journal, and the paper fulfils all the requirements to be published.
Author Response
Thanks for the positive comments of the reviewer and we will try our best to improve the quality of the manuscript.
Reviewer 3 Report
1- The author discusses in Table 2 the configuration of network parameters. Is it possible to adjust these parameters to enhance performance? 2- The comparison of PSNR and SSIM on data set5 in Table 3 needs to be clearer as it is difficult to comprehend all the results. It would be helpful to highlight the best technique by making the text bold. 3- The author has mentioned several algorithms, but has not explained the distinguishing factors of his proposed algorithm. 4- The authors discuss certain enhancements in comparison to other algorithms, citing improvements in terms of PSNR or SSIM. However, they only mention a slight improvement of 0.043 without providing information on execution time or other metrics. 5- The author has referred to two sets, namely Set5 and Set14. It would be beneficial if the author could clarify the differences between these sets and provide the results for all the other sets mentioned. 6- In Figure 7, the author discusses the average reconstruction time of various methods on data Set5 and data Set14. The author ran all algorithms on the same machine to ensure a fair comparison. 7- To improve clarity in terms of comparison, it is recommended to create a comprehensive table that includes all the different algorithms and their corresponding sets or parameters such as time, accuracy, SSIM, PSNR, etc. 8- To improve clarity in terms of comparison, it is recommended to create a comprehensive table that includes all different algorithms and their corresponding sets or parameters (such as time, accuracy, SSIM, PSNR, etc.) using various metrics. Additionally, for a fair comparison, all algorithms should be run on the same machine to assess their performance. 9- The process of reconstructing the image is unclear. 10- The contribution is unclear when compared to the related work. 11- The performance should be demonstrated for the overall dataset and not just for any specific image. It is recommended to present a table that includes all the datasets categorized by their differences. 12- More detailed information should be provided about the various data sets.Proofreading needed
Author Response
We had revised our manuscript carefully according to the reveiwer's suggestion and the detail response are listed as attached file.

Round 2
Reviewer 3 Report
The response is perfect.